# Chromatography-Independent Fractionation and Newly Identified Molecular Features of the Adzuki Bean (*Vigna angularis* Willd.) β-vignin Protein

**DOI:** 10.3390/ijms22063018

**Published:** 2021-03-16

**Authors:** Biane Philadelpho, Victória Souza, Fabiani Souza, Johnnie Santos, Fabiana Batista, Mariana Silva, Jessica Capraro, Stefano De Benedetti, Giuditta C. Heinzl, Eduardo Cilli, Alessio Scarafoni, Chiara Magni, Ederlan Ferreira

**Affiliations:** 1Department of Bromatological Analysis, School of Pharmacy, Federal University of Bahia, 40170-115 Salvador, Brazil; biane_philadelpho@hotmail.com (B.P.); victoriacruz.29@outlook.com (V.S.); fabipazelli@hotmail.com (F.S.); johnnie.machado25@gmail.com (J.S.); fabianaprb@gmail.com (F.B.); 2Chemistry Institute, Sao Paulo State University, 14800-900 Araraquara, Brazil; marianabarros.cs@gmail.com (M.S.); eduardocilli@gmail.com (E.C.); 3Department of Food, Environmental and Nutritional Sciences (DeFENS), Università degli Studi di Milano, 20133 Milan, Italy; jessica.capraro@unimi.it (J.C.); stefano.debenedetti@unimi.it (S.D.B.); giuditta.heinzl@unimi.it (G.C.H.); alessio.scarafoni@unimi.it (A.S.)

**Keywords:** protein vicilin-type, protein fractionation, biological activities, in vitro digestibility, amino acid sequencing, glycosylated polypeptides, metal binding capacity

## Abstract

Adzuki seed β-vignin, a vicilin-like globulin, has proven to exert various health-promoting biological activities, notably in cardiovascular health. A simple scalable enrichment procedure of this protein for further nutritional and functional studies is crucial. In this study, a simplified chromatography-independent protein fractionation procedure has been optimized and described. The electrophoretic analysis showed a high degree of homogeneity of β-vignin isolate. Furthermore, the molecular features of the purified protein were investigated. The adzuki bean β-vignin was found to have a native size of 146 kDa, and the molecular weight determined was consistent with a trimeric structure. These were identified in two main polypeptide chains (masses of 56–54 kDa) that are glycosylated polypeptides with metal binding capacity, and one minor polypeptide chain with a mass 37 kDa, wherein these features are absent. The in vitro analysis showed a high degree of digestibility of the protein (92%) and potential anti-inflammatory capacity. The results lay the basis not only for further investigation of the health-promoting properties of the adzuki bean β-vignin protein, but also for a possible application as nutraceutical molecule.

## 1. Introduction

Seed legumes are widely recognized nutritional sources of food that, among other sources, have gained attention for their high levels of protein. Grain legumes provide an alternative supply of protein in the diet. In this context, adzuki bean seeds are among the most common *Vigna* species used as human food [1]. Although the adzuki bean is native to the north-eastern part of China, it is cultivated and primarily consumed as a common food in many countries in east Asia, especially Japan, where it is also known as the red bean. The adzuki bean was introduced to southern United States, New Zealand, Latin America, parts of Africa, and European countries, where it is consumed as part of a vegetarian diet and as an alternative protein source to meat [2].

Adzuki bean, like other legumes, is considered an excellent source of high quality proteins and several other nutrients. The seeds of the *Vigna* species have been analysed and found to contain about 20–30% protein (dry basis). The amount found in the *V. angularis* species is approximately 21–28% [3,4,5]. The vicilin-like (7S) globulin accounts for about 80% of the total protein of the adzuki bean, whereas the legumin-like (11S) globulin makes up only about 10%. The latter consists of two types of subunits, the acidic and basic subunits with molecular masses of 40 kDa and 20 kDa, respectively [6]. Some varieties of adzuki bean contain only negligible amounts of the 11S globulin and albumins (2–8%). The 7S globulins of legume grains are heterogeneous [7], because of the uneven glycosylation of their subunits. The generally accepted structure of the 7S globulin is a trimer of polypeptides each with a Mr of 40 kDa to 75 kDa, thereby accounting for the Mr of 150 to 170 kDa in the native protein. 7S globulins generally do not contain cysteine amino acids, indicating the absence of disulphide bonds [8]. Indeed, Sakakibara et al. [9] described that the adzuki bean 7S globulin (β-vignin) consists of three subunits named α-, β’- and β-subunits with molecular weights of 55 kDa, 28 kDa, and 25 kDa, respectively. More recent studies, involving also molecular cloning of cDNAs, suggest that the major band (55 kDa) consists of at least two subunits of similar size [10].

In the last decade, health-promoting biological activities have been associated with several proteins from legume seeds, primarily soybean [11,12,13], chickpea [14,15], lupin [16], cowpea [5,17], adzuki [5], and mung [18] beans. Peptides of different sizes have been shown to exert various beneficial effects, including anticancer, antimicrobial, antioxidant, antihypertensive, cholesterol-lowering, immunomodulatory actions as well as obesity and Type 2 diabetes prevention [19,20,21,22]. 

Despite the rich protein content of the adzuki bean, the full potential of these beans has yet to be revealed, especially with respect to the nutraceutical properties of its components and of the protein fraction. Detailed investigation of the specific characteristics of legume seed storage proteins could lead to their better utilisation [23]. Thus, improved or simplified protocols for adzuki β-vignin protein fractionation and purification are actually required. Furthermore, the nutritional and nutraceutical properties of this protein have not yet thoroughly investigated. In this work, we describe a simplified enrichment procedure of the adzuki bean β-vignin protein. Its further molecular characterisation led us to identify novel molecular features, such as the real molecular weight of the native form, the glycosylation pattern, and the metal binding properties. Finally, functional properties relevant to nutritional issues, such as the in vitro digestibility and the potential anti-inflammatory capacity have been also investigated.

## 2. Results and Discussion

The analysis of chemical composition showed that the adzuki bean seed flour presented an average protein content of 204 g/kg (Table 1), an amount consistent with other reports [24]. Thus, adzuki bean seeds have gained popularity as nutritionally important sources of protein, similar to other seed legumes such as the soybean, lentil, and pea [1,17,25]. Approximately 81% of the total proteins present in the adzuki bean flour were solubilized and recovered by the fractionation technique used. The globulin fraction was predominant, accounting for 51.3% of total seed protein, followed by albumin (9.9%), glutelin (6.4%), and prolamin (2.5%) fractions. Only 3.9% of the total protein remained as an insoluble residue. This study is the first published work regarding the fractionation of the adzuki bean proteins. Globulins represent the major protein fraction (about 70%) of the TPE of the adzuki bean flour. Similarly, globulins are also the major protein type in other species such as cowpea flour (*Vigna unguiculata*), wherein they represent 42 and 62% of the TPE [26].

The solubility profile of a protein is an important characteristic, related not only to functional properties that affect the texture, colour, and sensory properties of products in which it is employed as an ingredient [27] but also for setting up effective purification procedures. 

The effect of pH on the protein solubility of adzuki bean globulin fraction in the absence of NaCl is displayed in Figure 1.

The results showed that maximum solubility occurred in very acidic (pH 1–2) or alkaline (pH 9–12) and a minimum between pH values of 4.0 and 5.0 the (around the isoelectric point value), which is typical solubility behavior of proteins from leguminous seeds. Various authors have reported this behavior at a pH below or above 4.0–4.5 and low ionic strength [8,27]. This value is in perfect agreement with that determined by the Protparam algorithm (available at www.expasy.org, accessed on 1 February 2021) using the three β-vignin amino acid sequences available, which calculated a pI of 4.4.

The purification procedures reported in the literature until now are not applicable for large-scale protein preparation [6,9,10]. Herein, a novel isolation procedure of the main protein fraction (β-vignin) is described (Figure 2) in which no chromatographic steps are required. The procedure is based on simple steps of differential solubilization of the proteins contained in the fraction extracted at pH 7.5 directly from the flour.

Thus, our protocol can be adopted for large-scale protein analyses in nutritional [14,18,22] and nutraceutical applicative fields [21], which may require animal or human testing [5,28], or for direct exploitation in specific formulation for human food [23]. 

Approximately 4 g of β-vignin protein was recovered from 100 g of adzuki bean flour, the degree of purity of the protein content of the freeze-dried powder was 96%. No yield optimization of the procedure was performed, since the industrialization of the processes is beyond the scope of this manuscript. 

Figure 3 showed the SDS-PAGE patterns of the isolated adzuki β-vignin and a highly purified β-vignin obtained chromatographic steps, as detailed in the Methods section, for purity comparison.

Gel images were analysed using the AlphaEase software; the adzuki bean TPE showed a major band with an estimated *M*r of 56.2 ± 2.8 kDa (59.4%). Several minor bands were observed with *M*r of 102.4 ± 2.4 kDa (4.7%), 87.0 ± 1.8 kDa (3.7%), 83.0 ± 2.8 kDa (5.3%), 68.3 ± 1.2 kDa (3.6%), 64.5 ± 2.2 kDa (7.0%), 37.0 kDa ± 2.1 (6.9%), 33.5 ± 1.1 kDa (5.6%), 29.6 ± 1.5 kDa (1.8%), and 20.7 ± 1.8 kDa (2.0%) in the TPE.

The bands corresponding to β-vignin represent the major polypeptide components of the extracts, accounting for over 70% of the total protein, while other polypeptides, likely mostly belonging to 11S globulin, represented only minor components.

Thus, the isolated β-vignin exhibited a high degree of homogeneity. Our results are likely driven by the overwhelming abundance of this protein in the studied seed and related to intrinsic protein composition, which could be attributed to cultivar differences, as well as to agronomic conditions related to sulfur availability [19]. 

Two-dimensional IEF/SDS-PAGE maps of the adzuki bean TPE and purified β-vignin are shown in Figure 4A,B, respectively. The 2D analysis allowed a higher level of resolution and, substantially, confirmed the high degree of homogeneity of the purified protein. The charge heterogeneity of the proteins is visible as a horizontal smear [3]. The experimentally determined pIs of the major globulin subunit ranged from 5.6 to 5.9. 

By and large, the obtained electrophoretic profiles displayed a lower degree of intrinsic heterogeneity and post-translational modifications relative to other seed storage globulins of this family, such as the lupin [7], pea [29], and soybean [30] 7S globulins. This is also supported by genes deposited in the databases for this protein, although there are only a limited number of sequences available, compared to other redundant gene families of homologous seed proteins.

However, charge heterogeneity due to minor sequence differences may be in question. In fact, ion-exchange chromatography separation clearly showed the presence of well-separated major and minor peaks (Data note shown), which suggests that preferential assortment of some subunits in the native trimers, namely the most acidic ones, might have occurred.

In order to confirm the identities of the purified protein, we carried out N-terminal amino acid sequencing of the adzuki bean β-vignin main spots that had been excised from the gel. This analysis showed all the same N-terminal residues, namely, isoleucine-valine-histidine-arginine (IVHR), corresponding to available sequencing of the mature polypeptides of the adzuki bean [28]. This finding confirmed the identity of the major subunit and that no N-terminal proteolytic processing event had occurred to the mature chains of the 7S globulin of the adzuki bean during their handling.

Gel molecular sieving of the β-vignin on a Sephadex^®^ G-200 column under native condition showed a unique protein peak (Figure 5).

The peak had an elution volume corresponding to an apparent molecular size of 146.4 ± 2.8 kDa for the adzuki bean 7S globulin. If the Mr values of the β-vignin principal constituent subunits determined by SDS-PAGE are considered, the oligomeric assembly of the adzuki bean β-vignin supports the conclusion that this *Vigna* seed globulin assembles into a trimeric structure, a common and widely acknowledged feature of most proteins of this class [31,32]. In the case of the adzuki bean 7S globulin, the resolved 3D structure (PDB 2EA7) leaves no doubt on the trimeric nature of these oligomers. This is unlike those found by Sakakibara et al. [9] in which the 7S protein showed one main band of 55 kDa and two minor protein bands of 28 kDa and 25 kDa. This was similarly observed by Meng and Ma [6]. Chen et al. (1984) demonstrated that the major seed protein of the Takara adzuki bean consists of two subunits with a relative mass Mr of 55 kDa and 35 kDa [3]. However, other studies suggested that β-vignin consists of three subunits with molecular weights of 55 kDa, 28 kDa, and 25 kDa [6,9].

To determine whether isoforms of similar Mr were present in the 55 kDa band, we performed SDS-PAGE runs under progressive sample dilutions. The main band consisted of two closely packed bands, the higher of which was more abundant (Figure 5, inset).

Next, we explored two molecular properties of β-vignin, i.e., the glycosylation pattern and the metal ion binding capacity. These properties may have relevant roles in protein structural stabilisation, protein-protein interactions, antibody recognition, metal transport, and other cellular events such as signalling processes and defence mechanisms [7].

The possible presence of N-acetylglucosamine-linked oligosaccharides in TPE and β-vignin was estimated by the reaction with lectin ConA (Figure 6B). The two main bands (56 kDa and 54 kDa) were significantly reactive, but the minor band (37 kDa) did not react with ConA. Thus, it can be concluded that the two main adzuki bean bands are alike in their glycosylation status, as both the polypeptides were glycosylated. In detail, the potential glycosylation site at residues 344–346 (NAT) was shared by all three adzuki sequences, thus confirming the N-glycosylation (Figure 7).

The amino acid sequence alignments of the adzuki bean β-vignin were conducted (Figure 7). The primary sequences of the adzuki bean 7S globulin (β-vignin) were obtained from NCBI and UniProtKB. The following entries were used: NCBI Blast: AB292246.1, UniProtKB: A4PI98_PHAAN (7S-1 subunit), A4PI99_PHAAN (7S-2 subunit), and A4PIA0_PHAAN (7S-3 subunit) for 7S adzuki globulin isoforms. The adzuki bean sequences showed a high degree of identity that ranged from 95% to 98%. The prediction of N-glycosylation was carried out by Net-N-Glyc, as available in the server at http://www.cbs.dtu.dk/services/NetNGlyc/ (accessed on 1 February 2021). Notably, the three adzuki bean sequences showed at least one N-glycosylation consensus sequence, namely, 344–346NAT, according to the adzuki bean sequence numbering. One of the three adzuki sequences also showed another potential glycosylation sequence at 89–91NGT. The amino acid sequences of soybean 7S globulins of the same family were used for comparative studies.

To test the metal interaction capability of the adzuki bean β-vignin protein, its binding in metal affinity chromatography (Ni-NTA) was monitored. For this purpose, the TPE of the adzuki bean were subjected to Ni-NTA column. The bound material was eluted with imidazole, and subjected to SDS-PAGE under reducing conditions. As can be seen in Figure 6A, the electrophoretic profiles of the Ni-NTA-bound material of the adzuki bean were a complete match. Therefore, this confirmed the metal binding capacity of the adzuki bean β-vignin. Similar results were shown with a cowpea bean β-vignin (*Vigna unguiculata* L. Walp.) [33]. 

In addition, the calcium, magnesium, nickel, and zinc contents of the adzuki bean isolated β-vignin were measured by ICP-MS. The ratios found in purified β-vignin are mostly negligible, except for the calcium and magnesium ions. Indeed, the molar ratios of each metal to protein monomer were 0.42, 0.15, 0.06, and 0.06, respectively. It cannot be excluded that the purification treatments caused the partial loss of the bound ions, which would give a very weak binding capacity. It is worth noting that the only 7S globulin of the *Vigna* species [34] that was found to be able to bind ions is that from the adzuki bean, which was shown to contain two calcium ion binding sites per subunit (PDB: 2EA7). This finding confirms the affinity of the protein to this metal ion. The present results confirmed that the globulin β-vignin is a metal ion binding protein, especially with respect to calcium ions. This feature is identical to that of the soybean β-conglycinin, in particular, the α’ subunits [11], but not with the lupin 7S canonical globulin [7]. It is possible that minor sequence differences or steric hindrance in the native globulin conformations give rise to this peculiar behaviour.

In vitro protein digestibility values of β-vignin, TPE and flour, from adzuki bean seed were lower than those of casein (Table 2). In particular, the flour displayed low digestibility (70.5%). However, the values of in vitro digestibility reported by Carbonaro et al. (1997) for the proteins of legumes such as bean (*Phaseolus vulgaris* L.), chickpea (*Cicer arietinum* L.), lentil (*Lens culinaris* Medikus), and faba bean (*Vicia faba* L.) were 74%, 78%, 82%, and 83%, respectively [25]. Han et al. (2007) reported digestibility values for the proteins of the species lentil (*Lens culinaris*), chickpea (*Cicer arietinum* L.), pea (*Pisum sativum* L.), and soybean (*Glycine max*) that were between 72% and 83% [35].

The greater digestibility of β-vignin protein (more than 92%) and TPE (84%) relative to the flour protein (*P <* 0.05) suggest the presence of other natural components in the seed that interact with proteins or enzymes and hinder the hydrolysis [23]. This is particularly applicable for the albumin fraction, which may contain seed protease inhibitors [26]. This would be expected in raw flours, but not in isolated and purified proteins from which anti-nutritional factors have been removed during their isolation procedure, as observed in our results (Table 2).

Finally, the possible anti-inflammatory property of purified β-vignin was investigated by assessing the modulation of chemokine IL-8 expression in cultivated Caco-2 cells. This cell line has been exploited for a range of studies aimed to elucidate the molecular mechanisms of food-derived compounds which may be difficult to address in vivo. Applications include the ability to elicit a reaction in response to pro-inflammatory stimuli [36]. In our case, inflammation was elicited using IL-1β, a pro-inflammatory cytokine that induces the expression of, among others, TNF-α, IL-6, and IL8 by triggering the NF-κB signaling pathway [33]. Cytokine IL-1 is able to stimulate IL-8 at both the mRNA and protein levels in Caco-2 cells [37]. It was observed in IL-1 stimulated undifferentiated Caco-2 cells that the effects of olive oil phenols on IL-8 mRNA mirrored those observed on IL-8 release [38].

Following incubation of β-vignin protein in the absence of IL-1β, NF-Kb activation was found comparable to that of control cells (Figure 8). Thus, the protein themselves was not able to stimulate inflammation responses. On the other hand, when the cells were stimulated by IL-1β, the presence of β-vignin caused a decrease of NF-kB activation of about 45%. 

## 3. Materials and Methods 

### 3.1. Plant Material and Reagents

Adzuki bean (*Vigna angularis* L.) seeds were kindly provided by the Centro Tecnológico da Zona da Mata da Empresa de Pesquisa Agropecuária de Minas Gerais (Viçosa, Brazil). The flour (not defatted) was prepared as previously described (Ferreira et al., 2018). All chemicals (purity ≥ 95%) were purchased from Sigma Aldrich^®^ (Saint Louis, MO, USA), unless stated otherwise.

### 3.2. Chemical Composition

The moisture content (drying at 105 °C to constant weight) and protein (total nitrogen, *N* × 5.70), lipid, ash (calcination of the sample in an oven at 550 °C), and total carbohydrate (by difference) compositions of the adzuki bean seed flour were determined [39].

### 3.3. Protein Solubility

The protein solubility of the adzuki bean seed flour was evaluated over a wide pH range (1.0–12.0). The total protein extract (TPE) was prepared by suspending 1 g of flour (ratio 1:30 *w/v*) in distilled water. The 2 M NaOH or HCl solutions were used for adjusting the pH. The slurry was then stirred for 30 min at room temperature, followed by centrifugation (5800 *g* for 30 min.). The supernatant was used to determine the concentration of soluble proteins [26]. The protein concentration was determined using bovine serum albumin as the reference [40].

### 3.4. Protein Fractionation and Isolation of Raw β-vignin from Adzuki Seed

The adzuki bean β-vignin protein was isolated according to the novel and simplified protocol shown in Figure 2. The dehulled adzuki bean seed flour was extracted twice with 0.5 M NaCl at pH 7.5 in the ratio 1:30 (*w/v*). The resulting supernatant was diluted 5 fold with distilled water and kept overnight at 4 °C after pH adjustment to 5.0 with HCl solution. Further centrifugation (12,000 *g* for 30 min) resulted in supernatant (albumins) and precipitate (globulins). The obtained pellet was washed with distilled water, extracted with 0.25 M NaCl at pH 5.3, and centrifuged resulting in precipitate (11S globulins) and supernatant containing 2S and 7S globulins. Dilution of the supernatant 5 fold (*v/v*) in distilled water, kept overnight at 4 °C after pH adjustment to 4.8 with HCl solution, allowed the separation of the two classes of protein. The pellet is constituted by isolated β-vignin (adzuki 7S globulin). 

The residue from the first NaCl extraction was extracted three times with ethanol (70%), and then with NaOH (1.0 M) to isolate the prolamin and glutelin fractions, respectively, from the final residue. The protein extracted at each step was determined by the Kjeldahl method using the conversion (*N ×* 5.7) value [39]. 

### 3.5. Molecular Exclusion Chromatography

The isolated adzuki bean β-vignin protein was fractionated by molecular exclusion chromatography as previously established [34], using a column packed with Sepharose^®^ CL-6B (100 × 2.5 cm) resin at a flow rate of 0.48 mL/min. The elution of the protein was monitored by measuring UV absorbance at 280 nm.

### 3.6. Ion-Exchange Chromatography

The adzuki bean β-vignin protein that had been purified by molecular exclusion chromatography was then further fractionated by ion-exchange chromatography on a Mono Q™ column (5.0 × 0.5 cm/cm), as previously established [34], using a linear gradient of NaCl (0.01–0.5 M) for 40 min at a flow rate of 1 mL/min, while monitoring at 280 nm.

### 3.7. Molecular Weight Determination

The apparent molecular mass was estimated by size exclusion chromatography on a Sephadex^®^ G-200 column (50 cm × 2.5 cm), as previously established [26], at a flow rate of 0.58 mL/min at room temperature. The standard proteins used for column calibration were ferritin (440 kDa), myosin (240 kDa), galactosidase (116 kDa), BSA (66 kDa), ovalbumin (45 kDa), and cytochrome C (12.4 kDa) (Sigma-Aldrich^®^, Saint Louis, MO, USA). The elution of proteins was monitored by measuring UV absorbance at 280 nm.

### 3.8. SDS-PAGE and IEF/SDS-PAGE

One-dimensional SDS-polyacrylamide gel electrophoresis (1D SDS-PAGE) was conducted using 12% polyacrylamide gels in the presence of 2% 2-mercaptoethanol [41] and a Hoefer MiniVE electrophoresis system (Amersham Biosciences^®^, Hercules, CA, USA). The following marker proteins of low molecular weight were used: rabbit muscle phosphorylase b (94 kDa), bovine serum albumin (66 kDa), hen egg white albumin (45 kDa), bovine carbonic anhydrase (29 kDa), soybean trypsin inhibitor (21.5 kDa), and hen egg white lysozyme (14.4 kDa) (GE Healthcare^®^, Little Chalfont, United Kingdom). Gel images were analysed using the AlphaEase software (Alpha Innotech^®^, San Leandro, USA).

Two-dimensional gel electrophoresis (2D IEF/SDS-PAGE) was performed on the TPE and the adzuki bean β-vignin. Adzuki bean flour (50 mg) was suspended in 1 mL of solution (7 M urea, containing 5 µg 1,4-dithiothreitol). The suspension was stirred for 30 min at room temperature and then centrifuged at 10,000 *g* for 10 min. The supernatant was diluted as follows: 20 μL of sample was added to 180 μL of redry solution (7 M urea, 2 M thiourea, 2% CHAPS, and 65 mM DTT). β-vignin protein was similarly prepared by adding 125 μL of redry solution to 20 μg of protein. Isoelectric focusing (IEF) was performed on 7 cm pH 3–10 nonlinear IPG strips, according to the manufacturer’s instructions using the Multiphor II electrophoresis unit (Amersham Biosciences^®^, Milan, Italy); the second dimension was then performed as described previously for 1D SDS-PAGE.

### 3.9. In Vitro Digestibility

The in vitro digestibility was assessed using pepsin (P-7012, Sigma-Aldrich^®^, Saint Louis, MO, USA) and pancreatin (P-7545, Sigma-Aldrich^®^, Saint Louis, MO, USA) in sequence, and incubated at 37 °C for 3 h and 24 h, respectively [42]. The reaction was stopped by adding 10% trichloroacetic acid, and then centrifuged (15,000 *g* for 15 min.). The amino-free groups in the supernatant were quantified by the reaction with 2,4,6-trinitrobenzenesulfonic acid (TNBS method) [43]. The amino acid L-leucine (0–100 nM) was used for the reference curve. The degree of hydrolysis (%DH) values were calculated as the percentage of free amino acids, expressed as micromoles of L-leucine, to the total micromoles of amino acids present in the original sample, using the following formula: %DH = [AAs − (AAba + AAbe)/AAtm] × 100, where *AAs* is the concentration of amino acid in the aliquot; *AAba* is the concentration of amino acid in the protein blank; *AAbe* is the concentration of amino acid in the enzyme black; and *AAtc* is the total concentration of amino acid in the aliquot. The in vitro digestibility was calculated by comparing the %DH of each sample to casein (C-8654, Sigma-Aldrich^®^, Saint Louis, MO, USA), which was used as the reference, under the same conditions. 

### 3.10. Detection of N-glycosylated Polypeptides by Concanavalin A (ConA)

Proteins separated by 1D SDS-PAGE were electrophoretically transferred to a nitrocellulose membrane using a mini Trans-blot Transfer Cell (Bio-Rad, Hercules, CA, USA). The blotted membranes were used to detect glycol-polypeptides by the ConA/peroxidase method [44].

### 3.11. N-Terminal Amino Acid Sequencing

After the proteins were separated by 1D SDS-PAGE, they were transferred to PVDF membranes by blotting on a Trans-blot Electrophoretic Transfer Cell (Bio-Rad, Milan, Italy), and the selected protein bands were excised and submitted for direct N-terminal amino acid sequencing. Automated Edman degradation was performed on a pulsed-liquid sequencer equipped with a PTH analyser (Procise model 491, Applied Biosystems^®^, Foster City, CA, USA).

### 3.12. Inductively Coupled Plasma/Mass Spectrometry (ICP-MS) for Globulin Metal Content

ICP-MS analysis (Aurorar^®^ M90 ICP-MS, Bruker, Bremen, Germany) was performed as previously established [45]. Briefly, the samples (about 0.2 g) were digested by a microwave digestion system (Multiwave-Eco^®^, Anton Paar, Rivoli, Italy) in Teflon^®^ tubes filled with 10 mL of 65% HNO_3_ by applying a two-step power ramp (Step 1: ramp to 200 W in 10 min, maintained for 5 min; Step 2: ramp to 650 W in 10 min, maintain for 15 min). In the mineralised sample, the concentrations of Mg, Ca, Ni, and Zn were measured. Typical analysis interferences were removed by using the collision-reaction interface at a H_2_ flow rate of 65 mL/min through the skimmer cone.

### 3.13. Immobilised Metal Affinity Chromatography

Immobilised metal affinity chromatography separation was carried out on a Ni-NTA-agarose (Qiagen^®^, Milan, Italy) column (1.8 cm × 5.0 cm). The TPE was prepared by suspending 2 g of flour (ratio 1:20 *w/v*) in (hydroxymethyl)aminomethane buffer (50 mM Tris-HCl, pH 7.5) containing NaCl (0.5 M). The slurries were stirred for 30 min at room temperature, and then centrifuged (12,000 *g* for 30 min.). Approximately 20 mg of the TPE was applied to the Ni-NTA-agarose column, which was equilibrated with the same buffer. The unbound material was removed by washing the column volume with the buffer at least five times. Thereafter, the bound material was eluted with the same buffer containing imidazole (0.1 M).

### 3.14. IL-8 Expression in Caco-2 Cells

Caco-2 cells were cultivated as previously described [46] and were seeded in 12-multiwell plates. The experiments were initiated on day 3 after cells reached confluence. Caco-2 cells were treated with 1 mg/mL of β-vignin for 1 h in the presence or absence of IL-1β (5 ng/mL) in the complete medium. Effects of the different molecules on inflammation were expressed as fold change in target genes expression relatively to the untreated control sample. qPCR was carried out as described in Capraro et al. [46]. Primers for amplification of IL-8 expressed gene were: 5-ATGACTTCCAAGCTGGCCGTGGCT-3 and 5-TCTCAGCCCTCTTCAAAAACTTCTC-3. The GAPDH reference gene was amplified with the primer pair: 5-GGAAGGTGAAGGTCGGAGTC-3 and 5-CACAAGCTTCCCGTTCTCAG-3. Each individual treatment was performed in triplicate.

### 3.15. Statistical Analysis

The means of the results were evaluated with one-way analysis of variance (ANOVA); for multiple comparison, Turkey’s test was used (SigmaStat^®^, v. 3.5, Systat software, CA, USA). The significance level was *P* ≤ 0.05. All results were expressed as mean ± standard deviation of at least three independent analyses.

## 4. Conclusions

In this study, a simplified enrichment procedure of the adzuki bean β-vignin protein with purity degree over 90% has been optimized and described. The method allows for large-scale protein preparation suitable for nutraceutical and other biotech applications.

In addition, data about the molecular features of β-vignin will enhance the information available for the adzuki bean protein fraction. Our molecular characterization showed that 81% of the total proteins present in the adzuki bean flour can be solubilized and recovered, of which the globulins represent the major fraction. The in vitro digestibility of the raw TPE and β-vignin was over 80% and 92%, respectively. β-vignin displayed a theoretical molecular size value of 147 kDa, according to densitometric analysis, which was confirmed by chromatographic techniques. The apparent Mr of the purified β-vignin was consistent with the widely acknowledged trimeric structure, which consists of two main bands (approximately 56 kDa and 54 kDa) that are both glycosylated polypeptide chains and one minor band of 37 kDa, in which this feature is absent. The protein presents a metal binding capacity. Interestingly, purified β-vignin showed anti-inflammatory properties when tested in cellular studies.

Finally, the information herein can lead to further investigation of the functional and nutritional properties of the adzuki bean β-vignin protein.

## Figures and Tables

**Figure 1 ijms-22-03018-f001:**
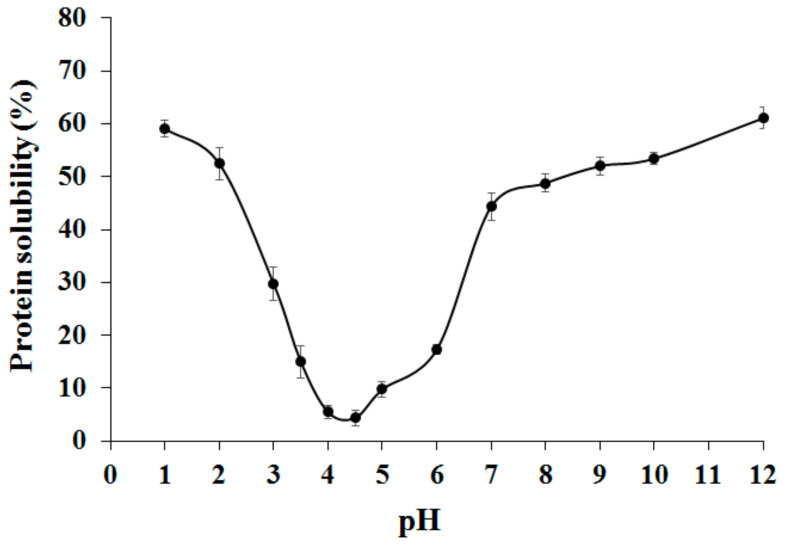
Effect of pH on adzuki bean globulins solubility.

**Figure 2 ijms-22-03018-f002:**
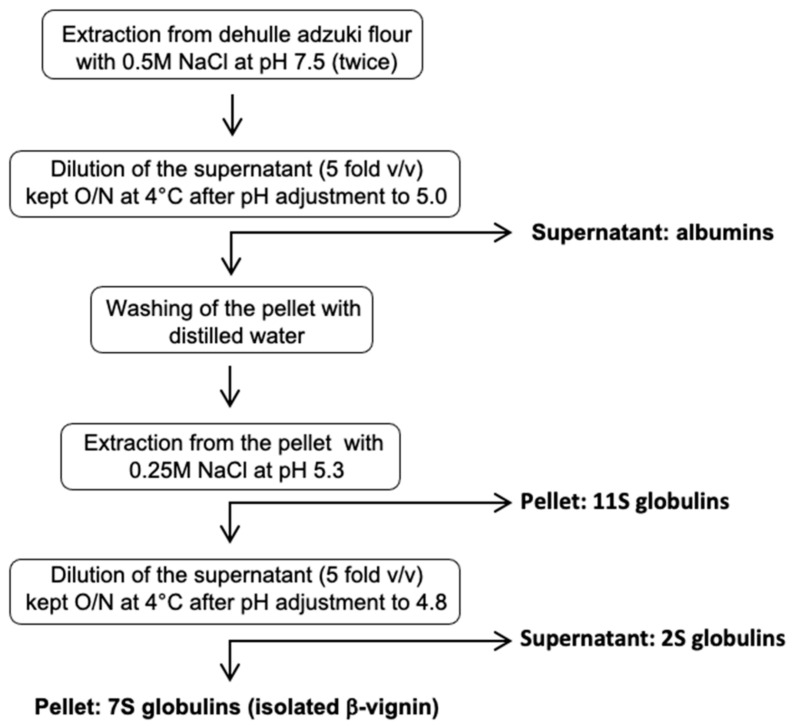
Schematic diagram of isolation of adzuki bean β-vignin protein by the preparative method.

**Figure 3 ijms-22-03018-f003:**
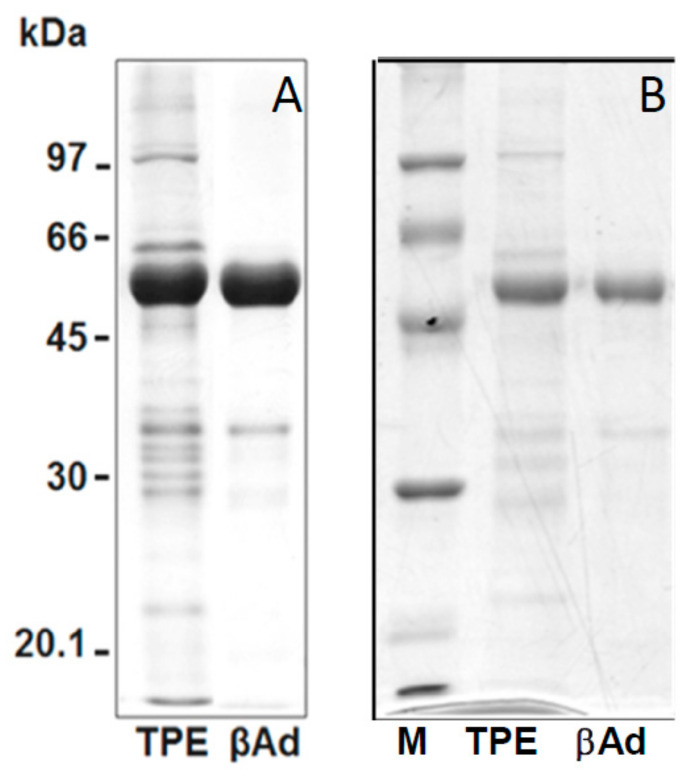
SDS-PAGE profile under reducing conditions of (**A**) isolated with the procedure detailed in Figure 2, and (**B**) chromatografically purified β-vignin. Molecular masses of protein markers are expressed as kDa. TPE represents the total protein extract from adzuki bean flour.

**Figure 4 ijms-22-03018-f004:**
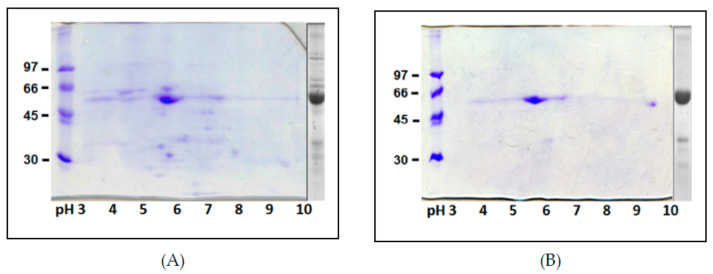
Two-dimensional IEF/SDS-PAGE of TPE (**A**) and highly purified β-vignin (**B**). The 1-D separation is reported to the right of each gel. Molecular masses of protein markers are expressed as kDa.

**Figure 5 ijms-22-03018-f005:**
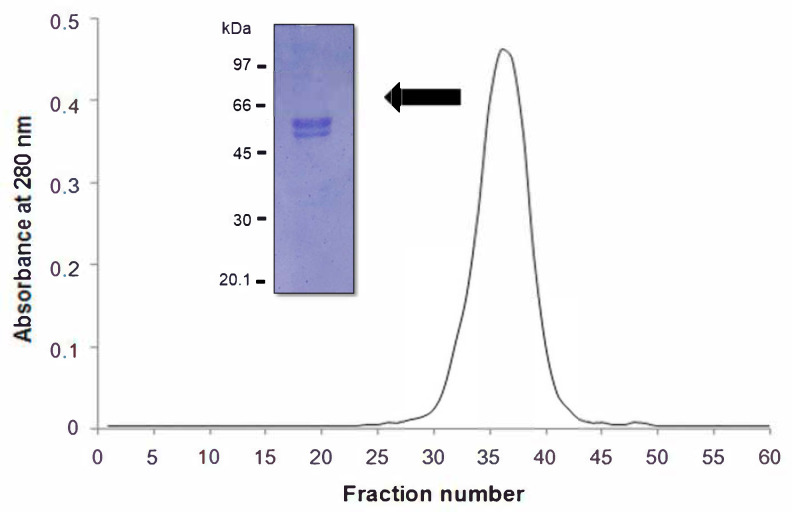
Gel molecular sieving on a Sephadex G-200 column of β-vignin. In the inset, SDS-PAGE profile under reducing conditions of purified β-vignin runs under progressive sample dilutions.

**Figure 6 ijms-22-03018-f006:**
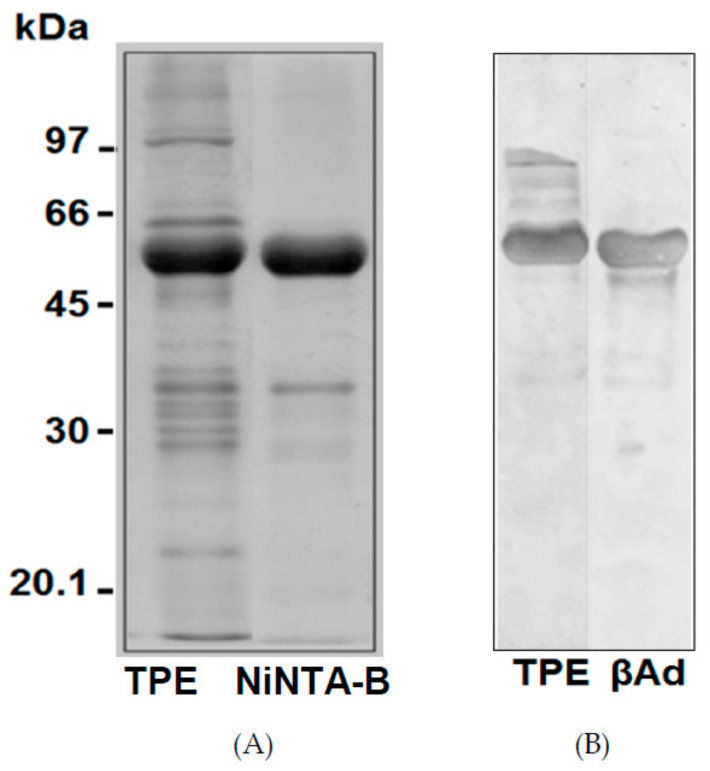
SDS-PAGE profile under reducing conditions of TPE and Ni-NTA-bound material (**A**), Western blot analysis with the concanavalin A (**B**). TPE represents the total protein extract from adzuki bean flour, βAd represents the adzuki bean β-vignin protein.

**Figure 7 ijms-22-03018-f007:**
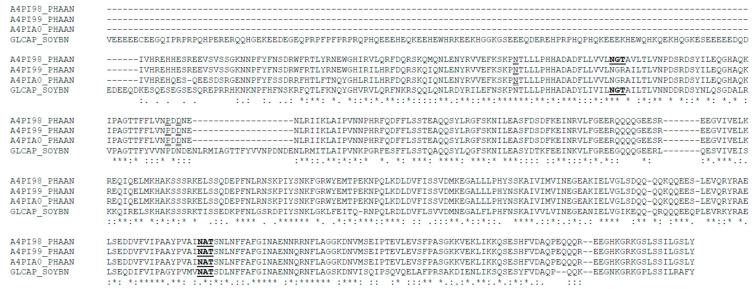
Adzuki bean β-vignin sequence alignment. The amino acid sequence of the β-vignin of adzuki (*Vigna angularis* Willd.), available at UniProt/TrEMBL, was aligned by using Clustal W 1.83 (PHAAN). The sequence of the soybean (SOYBN) homologous protein is also shown. Each complete line shows 100 amino acid residues. Asterisks: identity; semicolon: conserved substitution; full stop: semi-conserved substitution. N-glycosylation consensus sequences (NXS, T) are in bold and underlined. High prediction signal peptide (Signal P 4.0 Server) of adzuki 7S globulin, confirmed by N-terminal sequencing (see text), is italicized.

**Figure 8 ijms-22-03018-f008:**
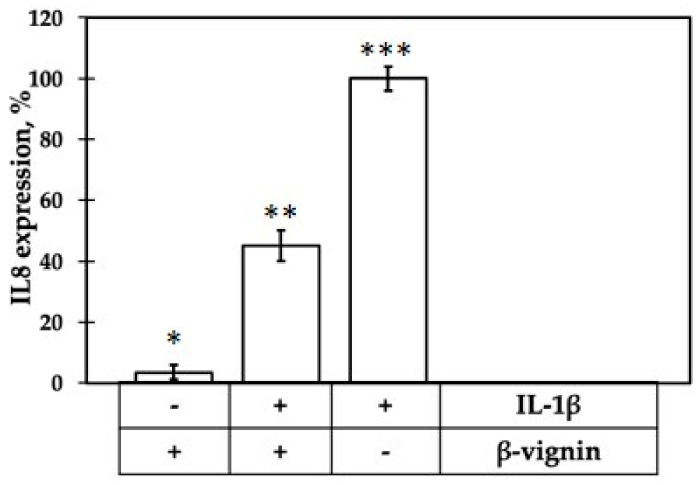
Inflammatory response of Caco-2 cells, assessed as IL-8 expression, elicited by treatment with IL-1β alone and incubated with β-vignin, in presence (+) or in absence (-) of IL-1β. Response to IL-1β alone was set as 100%. Mean value was significantly different: * *p* ≤ 0.05, ** *p* ≤ 0.01, *** *p* ≤ 0.001. See text for experimental details.

**Table 1 ijms-22-03018-t001:** Composition of the protein fraction of adzuki seed flour.

Fraction ^1^	Protein	
	mg/g of Seed	wt% ^2^
Wholemeal flour	203.90 ± 2.55	100
TPE	164.50 ± 1.87	80.68 ± 0.43
Albumin	20.22 ± 0.72	9.92 ± 0.26
Globulin	104.56 ± 1.34	51.28 ± 0.38
Prolamin	5.18 ± 0.12	2.54 ± 0.05
Glutelin	13.13 ± 0.32	6.44 ± 0.14
Insoluble protein	7.93 ± 0.42	3.89 ± 0.23

^1^ Values represent mean ± SD for three determinations. ^2^ wt% *N* × 5.70, quantified by the Kjeldahl method. For the calculation, 20.39% total protein in whole meal flour was used.

**Table 2 ijms-22-03018-t002:** In vitro digestibility of β-vignin protein, TPE, and flour from adzuki bean seed.

Sample *	Hydrolysis (%)	Digestibility (%) ^‡^
β-vignin	89.25 ± 2.09^b^	92.18 ± 1.73^b^
Total protein extract (TPE)	81.94 ± 1.66^c^	84.63 ± 0.92^c^
Flour	68.23 ± 2.02^d^	70.47 ± 1,84^d^
Casein (standard)	96.82 ± 0.82^a^	100.00 ± 1.91^a^

* Values represent means ± standard deviation of tests performed in triplicate. Different superscript letters mean statistical differences for *P* < 0.05 (ANOVA) by Turkey’s multiple-range test. ^‡^ Percentage relative to casein.

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
