# Peer review of "Chromatography-Independent Fractionation and Newly Identified Molecular Features of the Adzuki Bean (Vigna angularis Willd.) β-vignin Protein"

_ijms, 2021, doi:10.3390/ijms22063018_

Round 1
Reviewer 1 Report
- Line 17; “independent” spelled wrong
- Line 73-75, not sure what this means but seems nonsensical and unnecessary. Should be deleted or stated more directly.
- Still not addressed from last review> : Table 1, TPE in table under Wt% (change comma to decimal point).
- Methods 3.3., 3.4 How was pH adjusted; buffer or other (NaOH, HCL?), this is relevant to figure 1, figure 2, and methods section 3.3, and 3.4
- Line 128-129 needs correction for clarity.
- Line 159; to what does “This” refer? Homogeneity or is it being compared to other species where it is not a homogeneous?
- Figure 5. Elution position of markers not shown on figure. This makes it difficult to understand how mass estimations were obtained as stated in line 318-329
- Line 214-216; Better composition: To determine whether isoforms of similar Mr were present in the approx. 55 kDa band in figure 6, we performed SDS-PAGE runs ….
- Line 216; delete “Actually”
- not addressed from previous review, line 357 (now line 378), mL/tube is not a flow rate, give in mL/min.
- Line 394, Adzuzy ??? is Adzuki correct here?
Author Response
Response to reviewers
Firstly, we would like thanks the reviewers for the suggestions and considerations that will contribute to the scientific quality of the manuscript entitled “Chromatography-independent fractionation and newly identified molecular features of the adzuki bean (Vigna angularis Willd.) β-vignin protein”. All the criticisms and comments raised by the referee are answered below.
Reviewer 1 comments:
Point 1 – Line 17; “independent” spelled wrong.
Answer: The correct term has been included as requested by the review.
Point 2 – Line 73-75, not sure what this means but seems nonsensical and unnecessary. Should be deleted or stated more directly.
Answer: Adjusted, as requested by the reviewer.
Added in the text: “qPCR was carried out as described in Capraro et al. [46]. Primers for amplification of IL-8 expressed gene were: 5-ATGACTTCCAAGCTGGCCGTGGCT-3 and 5-TCTCAGCCCTCTTCAAAAACTTCTC-3. The GAPDH reference gene was amplified with the primer pair: 5-GGAAGGTGAAGGTCGGAGTC-3 and 5-CACAAGCTTCCCGTTCTCAG-3. Each individual treatment has been performed in triplicate”.
Point 3 – Still not addressed from last review: Table 1, TPE in table under Wt% (change comma to decimal point)
Answer: Adjusted, as requested by the reviewer.
Point 4 – Methods 3.3., 3.4 How was pH adjusted; buffer or other (NaOH, HCL?), this is relevant to figure 1, figure 2, and methods section 3.3, and 3.4
Answer: We agree with the reviewer. The requested details have been added to the relevant section of Methods (methods 3.3 and 3.4).
Point 6 – Line 128-129 needs correction for clarity.
Answer: The sentence has been rephrased, as requested by the review.
Added in the text: “Figure 3 showed the SDS-PAGE patterns of the isolated adzuki β-vignin anda highly purified β-vignin obtained chromatographic steps, as detailed in the Methods section, for purity comparison”.
Point 7 – Line 159; to what does “This” refer? Homogeneity or is it being compared to other species where it is not a homogeneous?
Answer: Clarified in the text, as requested by the review.
Added in the text: “Thus, the isolated β-vignin exhibited a high degree of homogeneity. Our results is likely driven by the overwhelming abundance of this protein in the studied seed and related to intrinsic protein composition, which could be attributed to cultivar differences, as well as to agronomic conditions related to sulfur availability [19]”.
Point 8 – Figure 5. Elution position of markers not shown on figure. This makes it difficult to understand how mass estimations were obtained as stated in line 318-329.
Answer: The calibration curve of the used chromatographic column has been included as Supplementary figure (S2).
Point 9 – Line 214-216; Better composition: To determine whether isoforms of similar Mr were present in the approx. 55 kDa band in figure 6, we performed SDS-PAGE runs
Answer: The sentence has been included in the text as requested by the review.
Added in the text: “To determine whether isoforms of similar Mr were present in the 55 kDa band
Point 10 – Line 216; delete “Actually”
Answer: Adjusted, as requested by the reviewer.
Point 11 – not addressed from previous review, line 357 (now line 378), mL/tube is not a flow rate, give in mL/min
Answer: The text has been corrected, as requested by the reviewer (flow rate of 0.58 mL/min).
Point 12 – Line 394, Adzuzy ??? is Adzuki correct here?
Answer: Adjusted, as requested by the reviewer.
We hope the revised version of the manuscript meets International Journal of Molecular Sciences publication criteria.
Sincerely.

Reviewer 2 Report
The authors have already published similar research "New molecular features of cowpea bean (Vigna unguiculata, l. Walp) β-vignin" in Biochemistry & Molecular Biology. I do not see the novelty on the isolation of beta- vignin protein but anti-inflammatory analysis may be novel data.
Figure 5 is not a high-resolution picture. It is better to replace it with HD picture to increase readers' reliability.
Line 743, There is no primer information on IL-1b gene expression measurement. There is no statistical analysis in Figure 8.
I doubt gene expression analysis because even though there is gene expression, sometimes there is no difference in protein level. The author should discuss protein level results as well with the current data.
Author Response
Response to reviewers
Firstly, we would like thanks the reviewers for the suggestions and considerations that will contribute to the scientific quality of the manuscript entitled “Chromatography-independent fractionation and newly identified molecular features of the adzuki bean (Vigna angularis Willd.) β-vignin protein”. All the criticisms and comments raised by the referee are answered below.
Reviewer 2 comment:
Point 1 – Figure 5 is not a high-resolution picture. It is better to replace it with HD picture to increase readers' reliability.
Answer: Adjusted, as requested by the reviewer.
Point 2 – Line 743, Ther is no primer information on IL-1b gene expression measurement. There is no statistical analysis in Figure 8.
Answer: The primers information was added (lines 455-459), as requested by the reviewer. Figure 8 has been modified, now reporting the statistical analysis.
Point 3 – I doubt gene expression analysis because even though there is gene expression, sometimes there is no difference in protein level. The author should discuss protein level results as well with the current data.
Answer: We understand the point. It was proved the same time course of IL-8 gene expression and protein level in stimulated Caco-2 cells. This issue is now discussed in the text (lines 300-303) and two references have been added to sustain the experimental choices.
Added in the text: “Cytokine IL-1 is able to stimulate IL-8 at both the mRNA and protein levels in Caco-2 cells [37]. It was observed in IL-1 stimulated undifferentiated Caco-2 cells that the effects of olive oil phenols on IL-8 mRNA mirrored those observed on IL-8 release [38]”
We hope the revised version of the manuscript meets International Journal of Molecular Sciences publication criteria.
Sincerely,
